# Polymorphisms in *ACE1*, *TMPRSS2*, *IFIH1*, *IFNAR2*, and *TYK2* Genes Are Associated with Worse Clinical Outcomes in COVID-19

**DOI:** 10.3390/genes14010029

**Published:** 2022-12-22

**Authors:** Cristine Dieter, Leticia de Almeida Brondani, Natália Emerim Lemos, Ariell Freires Schaeffer, Caroline Zanotto, Denise Taurino Ramos, Eliandra Girardi, Felipe Mateus Pellenz, Joiza Lins Camargo, Karla Suzana Moresco, Lucas Lima da Silva, Mariana Rauback Aubin, Mayara Souza de Oliveira, Tatiana Helena Rech, Luís Henrique Canani, Fernando Gerchman, Cristiane Bauermann Leitão, Daisy Crispim

**Affiliations:** 1Endocrine Division, Hospital de Clínicas de Porto Alegre, Porto Alegre 90035-903, RS, Brazil; 2Post-Graduate Program in Medical Sciences, Endocrinology, Department of Internal Medicine, Faculty of Medicine, Universidade Federal do Rio Grande do Sul, Porto Alegre 91501-970, RS, Brazil; 3Experimental Research Center, Hospital de Clínicas de Porto Alegre, Porto Alegre 90035-903, RS, Brazil; 4Diabetes and Metabolism Group, Centro de Pesquisa Clínica, Hospital de Clínicas de Porto Alegre, Porto Alegre 90035-903, RS, Brazil; 5Campus Realeza, Universidade Federal da Fronteira Sul, Realeza 85770-000, PR, Brazil

**Keywords:** polymorphisms, SARS-CoV-2, COVID-19, *ACE1*, *IFIH1*, *IFNAR2*, *TMPRSS2*, *TYK2*

## Abstract

Although advanced age, male sex, and some comorbidities impact the clinical course of COVID-19, these factors only partially explain the inter-individual variability in disease severity. Some studies have shown that genetic polymorphisms contribute to COVID-19 severity; however, the results are inconclusive. Thus, we investigated the association between polymorphisms in *ACE1*, *ACE2*, *DPP9*, *IFIH1*, *IFNAR2*, *IFNL4*, *TLR3*, *TMPRSS2*, and *TYK2* and the clinical course of COVID-19. A total of 694 patients with COVID-19 were categorized as: (1) ward inpatients (moderate symptoms) or patients admitted at the intensive care unit (ICU; severe symptoms); and (2) survivors or non-survivors. In females, the rs1990760/*IFIH1* T/T genotype was associated with risk of ICU admission and death. Moreover, the rs1799752/*ACE1* Ins and rs12329760/*TMPRSS2* T alleles were associated with risk of ICU admission. In non-white patients, the rs2236757/*IFNAR2* A/A genotype was associated with risk of ICU admission, while the rs1799752/*ACE1* Ins/Ins genotype, rs2236757/*IFNAR2* A/A genotype, and rs12329760/*TMPRSS2* T allele were associated with risk of death. Moreover, some of the analyzed polymorphisms interact in the risk of worse COVID-19 outcomes. In conclusion, this study shows an association of rs1799752/*ACE1*, rs1990760/*IFIH1*, rs2236757/*IFNAR2*, rs12329760/*TMPRSS2*, and rs2304256/*TYK2* polymorphisms with worse COVID-19 outcomes, especially among female and non-white patients.

## 1. Introduction

Coronavirus Disease 2019 (COVID-19) is a respiratory and systemic disease caused by the severe acute respiratory syndrome coronavirus 2 (SARS-CoV-2) [1]. According to the World Health Organization, 626 million people around the world have been infected by this virus, and 6,564,556 have died due to COVID-19 (https://covid19.who.int/ accessed on 25 October 2022). This disease is characterized by a variety of clinical manifestations ranging from asymptomatic to severe symptoms, which can progress to pneumonia, respiratory failure, multiple organ dysfunction, and death [2].

Although advanced age, male sex, obesity, diabetes mellitus (DM), and other comorbidities are associated with risk for the severe forms of the disease, these factors alone do not completely explain inter-individual variability in COVID-19 severity [2,3]. Therefore, the influence of genetic variations on clinical outcomes must be considered [3,4]. In this context, studies have reported the involvement of genetic polymorphisms in COVID-19 susceptibility and severity, being implicated in different biological pathways related to the disease [3,4].

Concerning COVID-19 susceptibility, the most studied polymorphisms are located in the *ACE2* and *TMPRSS2* genes, which are involved in viral binding and entry into host cells [5,6]. A common deletion/insertion (Del/Ins) polymorphism in intron 16 of the *ACE1* gene also seems to be associated with COVID-19, since the Del allele leads to decreased *ACE2* expression, which probably influences the process of virus entry into cells [7]. Moreover, a number of studies have indicated that polymorphisms in genes related to innate and adaptive immune response, such as *TLR* genes, are associated with COVID-19 development and/or severity [8]. Interferons (IFN) are canonical mediators of antiviral signaling that stimulate the release of many essential components of early host response to viral infection. Accordingly, polymorphisms in *IFN* genes or their receptors have been associated with increased susceptibility to COVID-19 or worse clinical outcomes [4]. Genome-wide association studies (GWAS) have described polymorphisms in other COVID-19 candidate genes, such as *DPP9* and *TYK2*, which have increased prevalence in COVID-19 patients [4]. The *DPP9* gene encodes a serine protease that has key immunomodulatory functions, including cleavage of the IFN-induced CXCL10 protein, regulation of T-lymphocyte proliferation, antigen presentation, and inflammasome activation [9,10,11]. The *TYK2* encodes a member of the Janus Kinase family of tyrosine kinases, which plays a key role in immune response against viral infections by mediating signaling pathways for type 1 IFN and several cytokines [4,12]. However, the associations of polymorphisms in these genes with COVID-19 have not been confirmed in different populations or investigated in Brazil.

Considering that genetic polymorphisms are probably involved in COVID-19 severity and there are differences between the frequencies of these polymorphisms across different populations and ethnicities, we investigated the association between the rs1799752/*ACE1*, rs2285666/*ACE2*, rs12329760/*TMPRSS2*, rs2109069/*DPP9*, rs2304256/*TYK2*, rs1990760/*IFIH1*, rs2236757/*IFNAR2*, rs368234815/*IFNL4*, and rs3775291/*TLR3* polymorphisms and COVID-19 severity and mortality in a Brazilian population.

## 2. Materials and Methods

### 2.1. Study Participants

Strengthening the Reporting of Observational studies in Epidemiology (STROBE) and Strengthening the Reporting of Genetic Association Studies (STREGA) guidelines were used to design and conduct this study [13,14]. The sample was a retrospective cohort of 694 unrelated COVID-19 patients selected from the Hospital de Clínicas de Porto Alegre–HCPA Biobank resource (COVID-19 Collection, (DOI:10.22491/hcpa-biobanco-amostras), application number: GPPG 2020-0218) [15]. Patients were recruited from this hospital between March and December 2020, before the national COVID-19 vaccine program began.

COVID-19 patients were categorized according to disease severity: 414 critically ill patients with severe COVID-19, admitted to the intensive care unit (ICU patients), and 280 non-critically ill patients referred to the hospital with mild or moderate symptoms and not admitted at the ICU (Ward inpatients). ICU group included patients with acute respiratory failure with the following criteria: respiratory frequency > 30 rpm, oxygen saturation < 90%, or the need for invasive or non-invasive mechanical ventilation [16]. Non-critically ill patients were those that did not fulfil the above criteria and were admitted to the medical ward, where they did not receive any kind of ventilation support besides supplemental oxygen. Patients were also classified as survivors (n = 469) or non-survivors (n = 183), which was defined as death during hospitalization.

Serum and plasma samples for biochemical analyses were taken on the first day after admission. Glycated hemoglobin (HbA1c) was analyzed with high performance liquid chromatography (HPLC) in BioRAD Variant Turbo II (BioRAD, Hercules, CA, USA). HbA1c values were expressed as percentages. DM was defined as previous diagnosis and/or HbA1c ≥ 6.5. Creatinine was measured by the Jaffe reaction (Sera-Pak immuno microalbuminuria, Bayer, Tarrytown, NY, USA; mean intra- and interassay coefficients of variance of 4.5% and 11%, respectively) [17]. Blood count was measured by light absorbance/impedance/flow cytometry; C-reactive protein (CRP) and D-dimers by immunoturbidimetry; sodium and potassium levels by indirect selective electrode method; lactate dehydrogenase (LDH) and urea by enzymatic methods; and lactate by amperometry. Ethnicity was defined based on self-classification, with patients being categorized as white or non-white.

The study protocol was approved by the Ethic Committee in Research from Hospital de Clínicas de Porto Alegre and, at the time of inclusion in the Biobank cohort, all patients or a next of kin provided assent and written informed consent prior to inclusion in the study.

### 2.2. SARS-CoV-2 Diagnostic Test

Nasopharyngeal swabs were collected from each patient and viral RNA was extracted from 600 μL respiratory species using the Abbott mSample Preparation System (Promega, Madison, WI, USA) in an Abbott M2000 Instrument (Abbott, Chicago, IL, USA). SARS-CoV-2 RNA positivity was evaluated by qualitative reverse transcription (RT)-polymerase chain reaction (PCR) in the Applied Biosystems QuantStudio Real-Time PCR 3 Instrument (Thermo Fisher Scientific, Waltham, MA, USA) by the Research Team of the Laboratório de Diagnóstico de SARS-CoV-2 of the Hospital de Clínicas de Porto Alegre, following a previously described protocol [18]. Patients were classified as SARS-CoV-2 positive if their cycle threshold (Ct) value was <40 for both nucleocapsid protein *N1* and *N2* genes, according to US Centers for Disease Control and Prevention RT-PCR protocol. The result was considered negative if the Ct was undetectable or >40, with adequate run control values. Three control samples were used in each RT-PCR run: a confirmed positive sample, a negative control (water), and an internal control (human *ribonuclease P* gene).

### 2.3. Genotyping

DNA was extracted from peripheral blood leucocytes using a FlexiGene DNA kit (Qiagen, Germantown, MD, USA). The rs1799752/*ACE1* (Del/Ins) (Assays ID = C_60538594A_10 and C_60538594B_20), rs2285666/*ACE2* (C/T) (Assay ID = C___2551626_1_), rs2109069/*DPP9* (G/A) (Assay ID = C__11517118_10), rs1990760/*IFIH1* (C/T) (Assay ID = C___2780299_30), rs2236757/*IFNAR2* (G/A) (Assay ID = C__11354003_30), rs368234815/*IFNL4* (TT/∆G) (Assay ID = C_203097338_10), rs3775291/*TLR3* (C/T) (Assay ID = C___1731425_10), rs12329760/*TMPRSS2* (C/T) (Assay ID = C__25622353_20), and rs2304256/*TYK2* (C/A) (Assay ID = C__25473911_10) polymorphisms were genotyped using specific Human TaqMan SNP Genotyping Assays 40× (Thermo Fisher Scientific). Detailed information about primer and probe sequences for each polymorphism can be consulted in the Thermo Fisher Scientific site (https://www.thermofisher.com/br/en/home.html accessed 28 November 2022) using the specific assay IDs. Real-time PCR reactions were performed in 384-well plates, in 5 µL total volume, using 2 ng of DNA, TaqPath ProAmp 1 × Mastermix (Thermo Fischer Scientific, Waltham, MA, USA) and TaqMan SNP Genotyping Assay 1×. Plates were placed in a RT-PCR thermal cycler (ViiA7 Real-Time PCR System; Thermo Fischer Scientific) and heated for 10 min at 95 °C, followed by 50 cycles of 95 °C for 15 s and 62 °C for 1 min. The rs1799752/*ACE1*, rs2285666/*ACE2*, rs12329760/*TMPRSS2*, rs2109069/*DPP9*, rs2304256/*TYK2*, rs1990760/*IFIH1*, rs2236757/*IFNAR2*, rs368234815/*IFNL4*, and rs3775291/*TLR3* polymorphisms were selected based on previous studies that have reported an association between these polymorphisms and COVID-19 susceptibility or severity or considering their functional effect in the expression of the respective genes (Table 1).

### 2.4. Statistical Analyses

Allele frequencies were calculated by gene counting and nonconformities with the Hardy–Weinberg equilibrium (HWE) were tested using χ^2^ tests. Allele and genotype frequencies were compared between groups of subjects using χ^2^ tests. Genotypes were also compared between groups under additive, recessive, and dominant inheritance models [57].

Clinical and laboratory characteristics were compared between patient groups using an unpaired Student’s *t*-test or χ^2^ test. For quantitative variables, the normality of distribution was determined using the Kolmogorov-Smirnov and Shapiro–Wilk tests. Variables with normal distribution are shown as mean ± SD. Variables with skewed distribution were log-transformed prior to analysis and are shown as median (25th–75th percentile values). Categorical variables are shown as n (%).

Multivariate logistic regressions were performed to evaluate the independent association of polymorphisms with COVID-19, adjusting for possible confounding factors. Variables that were significantly associated with COVID-19 in univariate analysis or biologically relevant were included in the multivariate model. Thus, multivariate analyses performed with the total sample included age, sex, and ethnicity as covariates. In the analyses stratified by sex, age and ethnicity were included in the multivariate model, while in the analyses stratified by ethnicity, age and sex were used as covariates. Statistical analyses were performed in IBM SPSS Statistics 18 (IBM, Armonk, NY, USA) with *p* values < 0.05 considered significant.

## 3. Results

### 3.1. Sample Description

The mean age of all patients was 59 ± 15.2 years (minimum 19 and maximum 97) and 47.1% of patients were males. Table 2 shows the clinical and laboratory characteristics of patients with COVID-19 disease categorized as inpatients or ICU patients. Males comprised 46.1% of the inpatient group and 57.5% of the ICU group (*p* = 0.004). The prevalence of DM and chronic obstructive pulmonary disease (COPD) was also higher in the ICU group compared to inpatients (all *p* < 0.050). Non-white patients were more frequent in the inpatient group (*p* = 0.006). As expected, percentages of death, need for invasive mechanical ventilation (IMV) and need for renal replacement therapy (RRT) were higher in ICU patients compared to inpatients (all *p* values < 0.050).

Table 3 describes the clinical and laboratory characteristics of COVID-19 patients categorized as survivors or non-survivors. The frequency of males and mean age were higher in non-survivors compared to survivors (*p* = 0.004 and *p* < 0.0001, respectively). Moreover, the prevalence of hypertension, DM, chronic kidney disease (CKD), cardiac failure, and cancer were higher in non-survivors than survivors (all *p* < 0.050). Need for IMV and presence of COPD were higher in non-survivors than survivors (all *p* < 0.0001).

### 3.2. Genotype and Allele Frequencies

Frequencies of the nine analyzed polymorphisms in patients with COVID-19 are described in Table 4 and Appendix A. The C/T genotype of rs2285666/*ACE2* was more frequent in the inpatient group than the ICU group (22.9% vs. 14.3%, *p* = 0.007); however, in the logistic regression analysis this association was not independent of age, sex, and ethnicity (OR = 1.173, 95% CI 0.731–1.882, *p* = 0.509; Appendix A). Both genotype and allele frequencies of the rs1799752/*ACE1*, rs2109069/*DPP9*, rs1990760/*IFIH1*, rs2236757/*IFNAR2*, rs368234815/*IFNL4*, rs3775291/*TLR3*, rs12329760/*TMPRSS2*, and rs2304256/*TYK2* polymorphisms did not differ significantly between the inpatient and ICU groups (all *p* > 0.050). No difference was found in the frequencies of these nine polymorphisms between survivors and non-survivors (all *p* > 0.050) (Table 4 and Appendix A).

### 3.3. Analyses after Stratification by Sex

Table 5 and Appendix A show frequencies of the analyzed polymorphisms in patients with COVID-19, stratified by sex. In males, no significant difference was found in the distributions of the nine polymorphisms of interest between inpatients and ICU patients or between survivors and non-survivors (all *p* > 0.050, Appendix A). In females, some of the analyzed polymorphisms were associated with COVID-19 outcomes (Table 5) and their results are detailed below.

The frequency of the rs1799752/*ACE1* Ins allele was higher in ICU patients compared to inpatients (41% vs. 32%, *p* = 0.018). The Ins/Ins genotype was also more frequent in ICU patients compared to inpatients (3.5% vs. 0.7%, *p* = 0.002), and the presence of the Ins allele (Del/Ins or Ins/Ins genotypes, dominant model) was associated with risk of severe COVID-19 (ICU admission), adjusted for age and ethnicity (OR = 2.283, 95% CI 1.366–3.814, *p* = 0.002). The frequency of the T allele of the rs1990760/*IFIH1* polymorphism was 46% in ICU patients and 37% in inpatients (*p* = 0.030). Moreover, the T/T genotype was associated with ICU risk in both recessive and additive models (OR = 1.855, 95% CI 1.024–3.359, *p* = 0.041, and OR = 2.153, 95% CI 1.097–4.226, *p* = 0.026, respectively), after adjustment for age and ethnicity. The frequency of the rs12329760/*TMPRSS2* T allele also differed between inpatients and ICU patients (15% vs. 22%, *p* = 0.021), and both the C/T and T/T genotypes conferred risk of ICU admission (OR = 1.717, 95% CI 1.065–2.769, *p* = 0.027, adjusting for covariables).

Interaction analysis between *ACE1*, *IFIH1* and *TMPRSS2* polymorphisms showed that having four, five, or six minor alleles of these polymorphisms increases the risk of ICU admission compared to carrying zero to three minor alleles (OR = 3.859, 95% CI 1.271–11.714, *p* = 0.017, after adjustment for age and ethnicity; Table 5). The other studied polymorphisms did not differ between ICU patients and inpatients in females (all *p* > 0.050; Appendix A).

When the nine polymorphisms were analyzed in females categorized accordingly to death occurrence, the frequency of the T/T genotype of the rs1990760/*IFIH1* polymorphism was 29% in non-survivors and 17.2% in survivors (*p* = 0.047, recessive model; Table 5). This genotype was associated with risk of death (OR = 2.281, 95% CI 1.175–4.427, *p* = 0.015), after adjustment for age and ethnicity. In addition, 40.8% of non-survivors carried the rs12329760/*TMPRSS2* T allele (C/T or T/T genotypes) and 31.7% of survivors (*p* = 0.196). After adjusting for age and ethnicity, the C/T and T/T genotypes appear to confer risk of death (OR = 1.647, 95% CI 0.927–2.927, *p* = 0.089, dominant model). The A/A genotype of the rs2304256/*TYK2* polymorphism showed a trend towards increased risk of death in the recessive model (OR = 2.574, 95% CI 0.882–7.513, *p* = 0.084, after adjustment for covariables; Table 5).

Although the associations of the rs12329760/*TMPRSS2* T allele and rs2304256/*TYK2* A/A genotype with death by COVID-19 did not reach statistical significance, interaction analysis between the *IFIH1*, *TMPRSS2*, and *TYK2* polymorphisms showed that the presence of four or more minor alleles was significantly associated with risk of death compared to 3 or fewer minor alleles (OR = 2.622, 95% CI 1.026–6.704, *p* = 0.044, after adjustment for age and ethnicity; Table 5). The other polymorphisms did not differ between females categorized as survivors or non-survivors (all *p* > 0.050, Appendix A).

### 3.4. Analyses after Stratification by Ethnicity

Table 6 and Appendix A show the frequencies of the analyzed polymorphisms in patients with COVID-19 after stratification by ethnicity. In white patients, no significant difference was found in the distributions of the nine polymorphisms between inpatients and ICU patients or between survivors and non-survivors (all *p* > 0.050, Appendix A). Significant associations among non-white patients are shown in Table 6, with the results detailed below.

The frequency of the A/A genotype of the rs2236757/*IFNAR2* polymorphism was 16.9% in ICU patients and 6.0% in inpatients (*p* = 0.045, recessive model). This genotype conferred risk of ICU admission in both recessive and additive models (OR = 3.238, 95% CI 1.116–9.391, *p* = 0.031, and OR = 3.375, 95% CI 1.116–10.208, *p* = 0.031, respectively), after adjustment for age and sex.

Moreover, the Ins/Ins genotype of the rs1799752/*ACE1* polymorphism was more frequent in non-survivors than survivors (7.1% vs. 0.0%, *p* = 0.011). This genotype was associated with risk of death both in the recessive (OR = 14.56, 95% CI 1.237–495.6, *p* = 0.032) and additive (OR = 14.98, 95% CI 1.188–537.5, *p* = 0.035) models, after adjustment for age and sex. The frequency of the rs2236757/*IFNAR2* A allele was 42.0% in non-survivors and 26.0% in survivors (*p* = 0.006). Accordingly, the A/A genotype of this polymorphism was present in 20.4% of non-survivors and 8.1% of survivors, being associated with risk of death after adjustment for covariables (OR = 3.682, 95% CI 1.213–11.176, *p* = 0.021). This association remained in both the additive and dominant models (*p* = 0.023 and *p* = 0.027, respectively; after adjustment for age and sex). Furthermore, the frequency of the rs12329760/*TMPRSS2* T allele was higher in non-survivors than survivors (27.0% vs. 17.0%, *p* = 0.042). Accordingly, the C/T and T/T genotypes (dominant model) were associated with risk of death (OR = 2.160, 95% CI 1.031–4.527, *p* = 0.041, after adjustment for age and sex).

Interaction analysis between *ACE1* and *TMPRSS2* polymorphisms showed that having three or four minor alleles increased the risk of death compared to two or fewer mutated alleles (OR = 5.854, 95% CI 1.280-26.777, *p* = 0.023, after adjustment for age and sex, Table 6). Risk of death was also increased in the presence of more than four minor alleles of the *ACE1*, *IFNAR2* and *TMPRSS2* polymorphisms (OR = 6.468, 95% CI 1.463–28.592, *p* = 0.014, after adjustment for the same covariables, Table 6).

## 4. Discussion

This study aimed to elucidate which genetic polymorphisms in nine candidate genes are associated with COVID-19 severity or mortality in a Brazilian population. We found that polymorphisms in *TMPRSS2*, *ACE1*, *IFNAR2*, and *IFIH1* genes are associated with worse clinical outcomes (ICU admission) and increased mortality in patients with COVID-19, and this is influenced by sex and ethnicity. Moreover, some of the analyzed polymorphisms interact in COVID-19 severity or mortality.

ACE2 receptor and TMPRSS2 are implicated in SARS-CoV-2 entry into human cells, which begins when the spike (S) viral protein binds to ACE2, followed by TMPRSS2 S-unit cleavage, resulting in the fusion of the virus membrane to the host cell membrane; which allows virus replication and spreading within the target cells [58]. Polymorphisms that influence the expressions of these genes could impact SARS-CoV-2 infection outcomes [59]. This hypothesis is supported by our results showing that the rs12329760/*TMPRSS2* T allele is associated with risk of ICU admission in females and death in non-white patients. Accordingly, Abdelsattar et al. [47] and Rokni et al. [48] reported that the rs12329760/*TMPRSS2* T allele was associated with COVID-19 severity in Egyptian (OR = 4.24, 95% CI 2.40–7.48) and Iranian (OR = 1.85, 95% CI 1.24–2.75) populations, respectively. However, in a meta-analysis of five studies, the T allele of this polymorphism conferred protection against severe forms of COVID-19 (OR = 0.77, 95% CI 0.66–0.91) [60]. These controversial results could be explained by ethnic differences among populations, interaction with other polymorphisms in each population, different risk factors, as well as sex, since we only observed this association in females. Regarding influence of sex, *TMPRSS2* expression is upregulated by androgens and has been associated with risk for prostate cancer, suggesting the predominance of COVID-19 infection in males could be partially explained by increased *TMPRSS2* expression in this sex [61,62]. Thus, it is plausible that the rs12329760/*TMPRSS2* T allele further increases *TMPRSS2* expression in females, increasing the odds of severe forms of COVID-19. In contrast, in males, the already high TMPRSS2 levels due to androgens may not be significantly impacted by the T allele.

Even though we observed no association between the rs2285666/*ACE2* polymorphism and COVID-19 outcomes, the presence of the Ins allele of the rs1799752/*ACE1* polymorphism conferred risk of ICU admission in females. Moreover, in non-white patients, the rs1799752/*ACE1* Ins/Ins genotype was associated with risk of death. ACE1 and ACE2 belong to the renin angiotensin aldosterone system (RAAS), which is involved in a variety of biological functions, including the regulation of blood pressure and inflammation, which are risk factors for COVID-19 [60,63]. ACE1 converts angiotensin (Ang) I to Ang II, promoting inflammation, thrombosis, and vasoconstriction, while ACE2 converts Ang II to Ang 1-7, counteracting most of the deleterious effects of Ang II activation. When SARS-CoV-2 binds to ACE2 receptor, it decreases ACE2 levels, thus disrupting the ACE1/ACE2 balance and causing RAAS activation, which may aggravate COVID-19 symptoms in the lungs, especially in patients with comorbidities [64,65]. The rs1799752/*ACE1* (Del/Ins) polymorphism is a 287-bp Alu insertion in intron 16 of the gene, which is associated with decreased ACE1 levels [66]. In a meta-analysis of five studies, Dieter et al. [60] found that the presence of the Ins allele conferred protection against severe forms of COVID-19. Two other meta-analyses [66,67] have also reported an association between the Ins allele and protection against severe forms of COVID-19, in disagreement with our results. Although the meta-analysis of Dobrijevic et al. [66] showed that the Ins/Ins genotype was associated with lower severity, it also showed that the presence of the Ins allele increased the risk of death due to COVID-19 in all analyzed inheritance models, which is in accordance with our results in non-white patients. The reasons for the conflicting results about the risk of COVID-19 severity and death related to the Ins allele are unknown at present, but could be due to heterogeneity of the studies, including the definition of COVID-19 severity, differences in the prevalence of comorbidities in each population, as well as ethnicity and sex differences. Unlike previous studies, which analyzed females and males together, we stratified our sample according to sex.

Type I IFN (IFN-I) is vital for immunity against viral infections [68]. In the context of COVID-19, some studies have reported that SARS-CoV-2 can inhibit IFN-I response in infected cells, leading to delayed response or overall suppression [69,70]. Thus, the virus can replicate and induce further tissue damage, triggering an exacerbated immune response, known as cytokine storm, as the immune system struggles to limit viral replication and to manage dying and dead cells [68]. The *IFNAR2* gene encodes a type I membrane protein that forms one of the two chains of the receptor for IFN α and β, being critical for IFN-I mediated immunity [71]. Here, we demonstrated an association between the rs2236757/*IFNAR2* A/A genotype and risk of both ICU admission and death in non-white patients. These results agree with Pairo-Castineira et al. [4], who reported an association between the rs2236757/*IFNAR2* A allele and risk of severe forms of COVID-19 in a UK population. Using Mendelian randomization, the authors also found evidence that lower expression of *IFNAR2* is associated with life-threating COVID-19 [4]. Hence, the rs2236757 polymorphism might decrease *IFNAR2* expression, consequently predisposing patients to severe forms of COVID-19.

The *IFIH1* gene encodes melanoma differentiation-associated protein 5 (MDA5) helicase receptor, which is a key intracellular sensor of viral double-stranded RNA that triggers the innate immune response, which leads to proinflammatory cascades, including the IFN-I response [36]. We reported an association between the T/T genotype of the rs1990760/*IFIH1* polymorphism and risk of ICU admission and death in females with COVID-19. In contrast, Minashkin et al. [36] found an association between the C/C genotype and risk of severe COVID-19. Of note, these authors used chest computer tomography to determine COVID-19 severity and compared healthy patients and patients with asymptomatic/mild/moderate symptoms to patients with severe pulmonary symptoms, unlike our population of patients with moderate and severe forms of the disease. Moreover, Amado-Rodríguez et al. [37] demonstrated that patients with COVID-19 and the rs1990760/*IFIH1* T/T genotype showed decreased levels of proinflammatory markers. Interestingly, most patients with the T/T genotype who were not treated with steroids survived the ICU stay. Conversely, patients with the T/T genotype treated with dexamethasone had a higher hospital mortality rate (HR = 2.19, 95% CI 1.01–4.87) and serum IL-6 levels [37], which is in line with our results. It is noteworthy that most of our patients were treated with steroids at hospital admission [41.8% of the inpatients vs. 71.4% of the ICU patients (*p* < 0.0001), and 52.6% of the survivors vs. 73.2% of non-survivors, (*p* < 0.0001)]. Including steroid treatment in the logistic regression analysis did not change the observed associations between the T/T genotype and COVID-19 severity or death in females. The rs1990760/*IFIH1* (C/T) polymorphism results in a change of an alanine to threonine (Ala946Tre) in MDA5, which predispose carriers to a number of autoimmune diseases [39]. In peripheral blood mononuclear cells, the 946Tre (T allele) causes increased IFN-I production. Accordingly, mice with the 646Tre variant show enhanced antiviral responses, being protected against viral infections [40]. Thus, taking mouse and cell cultures studies into account, it can be hypothesized that patients carrying the T allele (946Tre) might have higher IFN-I levels and a lower risk of COVID-19 [39]. However, as demonstrated here, other variables could influence this association in different populations, including steroid treatment and sex.

We also reported that interactions between polymorphisms in different genes might further increase the risk of more severe forms of COVID-19 or death. In females, a combination of the minor alleles of the rs1799752/*ACE1*, rs1990760/*IFIH1*, and rs12329760/*TMPRSS2* polymorphisms increased ICU admission risk, while a combination of rs1990760/*IFIH1*, rs12329760/*TMPRSS2*, and rs2304256/*TYK2* minor alleles increased the risk of death. Likewise, in non-white patients, interactions between the rs1799752/*ACE1* and rs12329760/*TMPRSS2* or rs1799752/*ACE1*, rs2236757/*IFNAR2*, and rs12329760/*TMPRSS2* minor alleles predisposed carriers to a higher risk of death than each individual polymorphism. Thus, the combined presence of risk alleles in genes related to viral entry into cells or immune response can lead to higher odds of progression to severe forms of COVID-19 or death.

As mentioned above, the effect of genetic risk factors on COVID-19 severity may diverge according to sex and ethnicity. Since the analyzed polymorphisms show variable frequencies across ethnicities and populations (https://www.ncbi.nlm.nih.gov/SNP accessed on 15 October 2022), we expected their effects to differ between white and non-white patients. Regarding sex, previous meta-analyses have shown that males have a higher risk of severe COVID-19 and death than females, probably due to hormonal influences and a higher prevalence of lifestyle-related risk factors, such as alcohol and smoking habits [72,73,74]. Moreover, some genes, such as *TMPRSS2*, are directly influenced by androgens. Until now, no study has analyzed polymorphism associations according to sex in patients with COVID-19. Thus, additional studies are necessary to confirm and clarify the differential effect of genetic polymorphisms on COVID-19 outcomes according to sex.

Although our results indicate the association between some genetic polymorphisms and COVID-19 severity, this study has certain limitations. First, we cannot exclude population stratification bias. To reduce the possibility of this bias, logistic regression analyses were performed with ethnicity and sex as covariates, and all patients were recruited from the same hospital. Second, we cannot fully exclude the possibility of a type II error when analyzing associations between polymorphisms and COVID-19 severity and death. Hence, for some polymorphisms that showed no association with COVID-19 outcomes, we cannot exclude the possibility that they could be associated with disease severity or death at lower ORs, which would require larger sample sizes to demonstrate the association. Third, besides sex and ethnicity, other risk factors are known to influence COVID-19 outcomes. Although the inclusion of DM and obesity in the logistic analyses did not significantly change the results, other unexplored factors could have biased our results. Unfortunately, due to the sample size and stratification by ethnicity and sex, we did not have adequate statistical power to take several factors into account.

## 5. Conclusions

Our study shows that the rs1799752/*ACE1*, rs1990760/*IFIH1*, rs2236757/*IFNAR2*, rs12329760/*TMRPSS2*, and rs2304256/*TYK2* polymorphisms are genetic risk factors for severe forms of COVID-19 and increased mortality. Further studies are necessary to confirm the association of these polymorphisms according to sex and different ethnicities. The finding of new genetic factors associated with COVID-19 progression might lead to new therapeutic avenues. Moreover, using genetic signatures to enrich COVID-19 prognostic could facilitate the screening of patients who are more likely to develop severe symptoms or to respond to a given therapy.

## Figures and Tables

**Table 1 genes-14-00029-t001:** Evidence about the nine polymorphisms analyzed in the present study.

Polymorphism–Gene	Localization	Previous Association with COVID-19	Functional Effect
rs1799752 (Del/Ins)—*Angiotensin I converting enzyme* (*ACE1*)	Chr 17—intron	- The Del allele was associated with risk of COVID-19 in different populations [19,20,21,22,23,24]. - The Ins/Ins genotype was associated with protection against COVID-19 [25]. - The Ins/Ins genotype was associated with risk of COVID-19 [26]. - No association with COVID-19 susceptibility or severity [27,28,29]. - *ACE1* expression was higher in ICU patients than non-ICU patients. Patients carrying the Del/Ins genotype had a higher *ACE1* expression than Ins/Ins genotype carriers [30].	The Del/Del genotype increased *ACE1* expression [31].
rs2285666 (C/T)—*Angiotensin converting enzyme 2* (*ACE2*)	Chr X—intron	- The T allele was more frequent in symptomatic patients vs. asymptomatic (19). - No association with COVID-19 susceptibility or severity [21,27,29]. - The C/C genotype was associated with risk of COVID-19 susceptibility and/or death [28,32,33]. - *ACE2* expression did not differ between COVID-19 patients and healthy controls [30].	The T allele increased *ACE2* expression [34,35].
rs2109069 (G/A)—*Dipeptidyl peptidase 9* (*DPP9*)	Chr 19—intron	- Association with COVID-19 severity [4].	Information not available.
rs1990760 (C/T)—*Interferon induced with helicase C domain 1* (*IFIH1*)	Chr 2—exon	- The C/C genotype was associated with risk of COVID-19 severity [36]. - *IFIH1* expression was lower in COVID-19 patients carrying the T/T genotype [37].	The T allele increases *IFIH1* expression [38]. The T allele increased IFN-I production [39] and protected mice against viral infections [40].
rs2236757 (G/A)—*Interferon α and β receptor subunit 2* (*IFNAR2*)	Chr 21—intron	- The A allele was associated with risk of severe COVID-19 [4]. - *IFNAR2* lower expression was associated with life-threating COVID-19.	Information not available.
rs368234815 (TT/∆G)—Interferon lambda 4 (*IFNL4*)	Chr19—exon	- The ΔG/ΔG genotype was associated with risk of higher viral loads in COVID-19 patients [41]. - The ΔG/ΔG genotype was associated with risk of severe COVID-19 [42].	This polymorphism does not affect *IFNL4* mRNA expression levels, but the IFNL4 protein only is produced in the presence of the ΔG allele [43].
rs3775291 (C/T)—*Toll like receptor 3* (*TLR3*)	Chr 4—exon	- The T allele predisposed to SARS-CoV2 infection and associated mortality [44]. - No association with COVID-19 susceptibility and mortality [45].	This polymorphism does not seem to directly affect *TLR3* expression [46].
rs12329760 (C/T)—Transmembrane serine protease 2 (*TMPRSS2*)	Chr 21—exon	- The T allele was associated with COVID-19 severity [47,48]. - The T allele was more frequent among mild cases of COVID-19 than severe cases [49,50,51]. - The T allele was associated with protection against COVID-19 [52]. - No association with COVID-19 susceptibility and mortality [53,54].	In silico analysis showed that this polymorphism might have an impact at the *TMPRSS2* mRNA level [55]. The T allele increased *TMPRSS2* expression [53].
rs2304256 (C/A) —*Tyrosine kinase 2* (*TYK2*)	Chr 19—exon	- The rs74956615 polymorphism near the gene that encodes TYK2 was associated with COVID-19 severity [4]. - No previous data about the rs2304256/*TYK2* polymorphism in COVID-19 patients.	The A allele decreased *TYK2* expression [56].

**Table 2 genes-14-00029-t002:** Clinical and laboratory characteristics of patients with COVID-19 categorized into inpatients and ICU patients.

Characteristics	All Patients (n = 694)	Inpatients (n = 280)	ICU Patients (n = 414)	*p* Values *
Age (years)	59 ± 15.2	57 ± 17	59.5 ± 13.8	0.082
Sex (% males)	47.1	46.1	57.5	0.004
Ethnicity (% non-whites)	25.9	31.8	22.1	0.006
BMI (kg/m^2^)	30.4 ± 7.1	29.2 ± 7.3	30.9 ± 7.0	0.009
Severity				
Death (%)	26.3	7.5	39.1	<0.0001
Need for IMV (%)	46.3	0	77.3	-
Need for RRT (%)	21.1	5.4	31.8	<0.0001
Comorbidities				
Hypertension (%)	72.1	73.8	71.1	0.565
Obesity (%)	46.6	43.5	48.1	0.325
DM (%)	44.6	35.2	50.5	<0.0001
Chronic kidney disease (%)	19.9	19.0	20.5	0.719
Cancer (%)	11.1	16.3	7.8	0.001
Cardiac failure (%)	15.2	15.9	14.8	0.802
Asthma (%)	11.6	10.5	12.2	0.677
COPD (%)	7.0	4.2	8.8	0.035
Organ transplantation (%)	5.0	6.8	3.9	0.131
Blood counts				
Hematocrit (%)	36.6 ± 6.3	37.2 ± 6.2	36.1 ± 6.4	0.031
Hemoglobin (g/dL)	12.2 ± 2.3	12.4 ± 2.2	12.0 ± 2.3	0.017
Leukocytes (×10^3^/μL)	8.14 (5.74–11.79)	7.26 (5.43–10.03)	9.07 (6.21–13.21)	<0.0001
Lymphocytes (%)	10.4 (6.2–15.9)	14.4 (9.9–23.2)	8.3 (5.0–12.5)	<0.0001
Platelets (×10^3^/μL)	221.0 (164.0–272.0)	211.0 (157.3–275.8)	225.0 (167.0–272.0)	0.544
Creatinine (mg/dL)	0.98 (0.77–1.61)	0.90 (0.74–1.19)	1.05 (0.78–1.80)	<0.0001
eGFR (mL/min per 1.73m^2^)	73.0 (40.1–91.0)	79.0 (54.0–94.0)	65.0 (33.2–90.0)	<0.0001
Sodium (mEq/L)	138.9 ± 4.7	138.1 ± 4.2	139.5 ± 4.9	<0.0001
Potassium (mEq/L)	4.3 ± 0.7	4.2 ± 0.6	4.4 ± 0.7	0.014
C-reactive protein (mg/L)	116.0 (49.5–186.9)	69.3 (25.7–149.7)	134.2 (79.3–211.1)	<0.0001
Urea (mg/L)	43.0 (31.0–72.8)	37.0 (27.0–53.0)	50.0 (34.0–81.0)	<0.0001
D-dimer (μg/mL)	1.16 (0.60–2.42)	0.81 (0.5–1.48)	1.46 (0.68–3.54)	<0.0001
Lactate (mmol/L)	1.30 (0.96–1.70)	1.14 (0.90–1.64)	1.3 (1.0–1.7)	0.006
LDH (U/L)	375.0 (268.0–526.0)	281.5 (220.8–375.0)	455.0 (348.0–608.0)	<0.0001
HbA1c (%)	6.0 (5.5–7.1)	5.8 (5.3–6.5)	6.1 (5.6–7.4)	<0.0001

Variables are shown as mean ± SD, median (25th–75th percentiles) or percent. * *p* value was computed using Student’s *t* tests, Mann–Whitney U tests or χ^2^ tests, as appropriate. BMI: body mass index; COPD: chronic obstructive pulmonary disease; DM: diabetes mellitus; eGFR: estimated glomerular filtration rate; HbA1c: glycated hemoglobin; IMV: invasive mechanical ventilation; LDH: lactate dehydrogenase; RRT: renal replacement therapy.

**Table 3 genes-14-00029-t003:** Clinical and laboratory characteristics of patients with COVID-19 categorized into survivors and non-survivors.

Characteristics	Survivors (n = 469)	Non-Survivors (n = 183)	*p* Values *
Age (years)	55 ± 15	65 ± 13	<0.0001
Sex (% males)	48.4	61.7	0.004
Ethnicity (% non-whites)	27.7	24.3	0.447
BMI (kg/m^2^)	30.8 ± 7.1	29.2 ± 6.5	0.011
Severity			
Need for ICU admission (%)	32.1	98.4	<0.0001
Need for IMV (%)	31.5	84.2	<0.0001
Need for RRT (%)	11.9	46.4	<0.0001
Comorbidities			
Hypertension (%)	68.4	79.5	0.012
Obesity (%)	48.1	40.6	0.126
DM (%)	39.4	55.2	0.001
Chronic kidney disease (%)	16.7	30.6	<0.0001
Cancer (%)	9.8	16.0	0.037
Cardiac failure (%)	14.4	18.4	0.311
COPD (%)	4.2	14.0	<0.0001
Organ transplantation (%)	5.5	5.0	0.940
Blood counts			
Hematocrit (%)	37.0 ± 5.9	35.2 ± 6.9	0.002
Hemoglobin (g/dL)	12.4 ± 2.2	11.6 ± 2.4	<0.0001
Leukocytes (×10^3^/μL)	7.76 (5.58–11.06)	9.24 (6.19–15.10)	0.252
Lymphocytes (%)	12.0 (7.83–18.2)	7.2 (5.0–12.5)	<0.0001
Platelets (×10^3^/μL)	226.0 (166.7–272.2)	212.5 (144.8–271.3)	0.063
Creatinine (mg/dL)	0.91 (0.74–1.26)	1.42 (0.81–2.25)	<0.0001
eGFR (mL/min per 1.73m^2^)	78.0 (51.0–94.0)	47.0 (24.0–85.0)	<0.0001
Sodium (mEq/L)	138.8 ± 4.1	139.3 ± 5.5	0.238
Potassium (mEq/L)	4.2 ± 0.6	4.5 ± 0.8	<0.0001
C-reactive protein (mg/L)	99.7 (39.9–173.3)	136.7 (73.3–207.8)	<0.0001
Urea (mg/L)	37.0 (29.0–55.7)	67.0 (43.0–99.0)	<0.0001
D-dimer (μg/mL)	0.93 (0.54–1.94)	1.72 (0.86–3.75)	<0.0001
Lactate (mmol/L)	1.20 (0.90–1.60)	1.30 (1.0–1.70)	0.006
LDH (U/L)	356.0 (244.0–480.5)	481.0 (325.0–638.2)	<0.0001
HbA1c (%)	5.8 (5.4–6.7)	6.3 (5.7–7.4)	0.037

Variables are shown as mean ± SD, median (25th–75th percentiles) or percent. * *p* value was computed using Student’s *t* tests, Mann–Whitney U tests or χ^2^ tests, as appropriate. BMI: body mass index; COPD: chronic obstructive pulmonary disease; DM: diabetes mellitus; eGFR: estimated glomerular filtration rate; HbA1c: glycated hemoglobin; IMV: invasive mechanical ventilation; LDH: lactate dehydrogenase; RRT: renal replacement therapy.

**Table 4 genes-14-00029-t004:** Genotype and allele frequencies of the analyzed polymorphisms in patients with COVID-19.

	Inpatients	ICU Patients	*p* *	Survivors	Non-Survivors	*p* *	Test for HWE *p* *
rs1799752—*ACE1*	277	403		463	176		
Genotype							
Del/Del	85 (30.7)	113 (28.0)	0.138	132 (28.5)	49 (27.8)	0.266	<0.00001
Del/Ins	188 (67.9)	274 (68.0)		321 (69.3)	119 (67.6)		
Ins/Ins	4 (1.4)	16 (4.0)		10 (2.2)	8 (4.6)		
Allele							
Del	0.65	0.62	0.360	0.63	0.62	0.660	-
Ins	0.35	0.38		0.37	0.38		
rs2285666—*ACE2*	279	406		463	180		
Genotype							
C/C	182 (65.2)	281 (69.2)	0.007	308 (66.5)	131 (72.8)	0.222	<0.00001
C/T	64 (22.9)	58 (14.3)		93 (20.1)	26 (14.4)		
T/T	33 (11.9)	67 (16.5)		62 (13.4)	23 (12.8)		
Allele							
C	0.77	0.76	0.932	0.77	0.80	0.211	-
T	0.23	0.24		0.23	0.20		
rs2109069—*DPP9*	280	411		467	182		
Genotype							
G/G	114 (51.4)	202 (49.1)	0.376	241 (51.6)	87 (47.8)	0.591	0.905
G/A	116 (41.4)	167 (40.6)		191 (40.9)	78 (42.9)		
A/A	20 (7.2)	42 (10.3)		35 (7.5)	17 (9.3)		
Allele							
G	0.69	0.69	0.847	0.72	0.69	0.346	-
A	0.31	0.31		0.28	0.31		
rs1990760—*IFIH1*	274	401		461	174		
Genotype							
C/C	93 (33.9)	122 (30.4)	0.481	147 (31.8)	60 (34.5)	0.255	0.488
C/T	131 (47.8)	193 (48.2)		228 (49.5)	74 (42.5)		
T/T	50 (18.3)	86 (21.4)		86 (18.7)	40 (23.0)		
Allele							
C	0.58	0.54	0.245	0.57	0.56	0.829	-
T	0.42	0.46		0.43	0.44		
rs223675—*IFNAR2*	279	410		466	181		
Genotype							
G/G	124 (44.4)	174 (42.4)	0.740	212 (45.5)	73 (40.3)	0.356	0.638
G/A	119 (42.7)	187 (45.6)		195 (41.8)	87 (48.1)		
A/A	36 (12.9)	49 (12.0)		59 (12.7)	21 (11.6)		
Allele							
G	0.66	0.65	0.885	0.66	0.64	0.526	-
A	0.34	0.35		0.34	0.36		
rs368234815—*IFNL4*	274	404		460	177		
Genotype							
TT/TT	110 (40.1)	151 (37.4)	0.687	170 (37.0)	72 (40.7)	0.580	0.007
TT/∆G	117 (42.7)	175 (43.3)		201 (43.7)	76 (42.9)		
∆G/∆G	47 (17.2)	78 (19.3)		89 (19.3)	29 (16.4)		
Allele							
TT	0.61	0.59	0.394	0.59	0.62	0.306	-
G	0.39	0.41		0.41	0.38		
rs3775291—*TLR3*	278	404		463	179		
Genotype							
C/C	136 (48.9)	183 (45.3)	0.574	224 (48.4)	79 (44.1)	0.562	0.642
C/T	112 (40.3)	179 (44.3)		194 (41.9)	79 (44.1)		
T/T	30 (10.8)	42 (10.4)		45 (9.7)	21 (11.8)		
Allele							
C	0.69	0.67	0.569	0.69	0.66	0.312	-
T	0.31	0.33		0.31	0.34		
rs12329760—*TMPRSS2*	279	410		465	182		
Genotype							
C/C	192 (68.8)	263 (64.2)	0.316	312 (67.1)	115 (63.2)	0.633	0.654
C/T	77 (27.6)	135 (32.9)		138 (29.7)	60 (33.0)		
T/T	10 (3.6)	12 (2.9)		15 (3.2)	7 (3.8)		
Allele							
C	0.83	0.81	0.384	0.82	0.80	0.389	-
T	0.17	0.19		0.18	0.20		
rs2304256—*TYK2*	279	408		463	182		
Genotype							
C/C	161 (57.7)	225 (55.1)	0.426	261 (56.4)	101 (55.5)	0.272	0.272
C/A	107 (38.4)	158 (38.7)		183 (39.5)	68 (37.4)		
T/A	11 (3.9)	25 (6.2)		19 (4.1)	13 (7.1)		
Allele							
C	0.77	0.74	0.347	0.76	0.74	0.506	-
A	0.23	0.26		0.24	0.26		

Data are shown as number (%) or proportion. * *p*-values were calculated using χ^2^ tests. HWE = Hardy–Weinberg Equilibrium.

**Table 5 genes-14-00029-t005:** Genotype and allele frequencies of the analyzed polymorphisms in females with COVID-19 according to disease severity and mortality.

Sex	Groups		Unadjusted *p* *	Adjusted OR (95% IC)/*p* ^†^
Females	Inpatients	ICU Patients		
rs1799752/*ACE1*	149	171		
Genotype				
Del/Del	55 (36.9)	36 (21.1)	0.002	1
Del/Ins	93 (62.4)	129 (75.4)		2.203 (1.316–3.688)/0.003
Ins/Ins	1 (0.7)	6 (3.5)		8.127 (0.932–70.843)/0.058
Allele				
Del	0.68	0.59	0.018	-
Ins	0.32	0.41		
Dominant model				
Del/Del	55 (36.9)	36 (21.1)	0.003	1
Del/Ins + Ins/Ins	94 (63.1)	135 (78.9)		2.283 (1.366–3.814)/0.002
rs1990760–*IFIH1*	147	172		
Genotype				
C/C	59 (40.1)	55 (32.0)	0.081	1
C/T	66 (44.9)	75 (43.6)		1.281 (0.762–2.155)/0.350
T/T	22 (15.0)	42 (24.4)		2.139 (1.100–4.161)/0.025
Allele				
C	0.63	0.54	0.030	-
T	0.37	0.46		
Recessive model				
C/C+ C/T	125 (85.0)	130 (75.6)	0.050	1
T/T	22 (15.0)	42 (24.4)		1.855 (1.024–3.359)/0.041
Additive model				
C/C	59 (72.8)	55 (56.7)	0.038	1
T/T	22 (27.2)	42 (43.3)		2.153 (1.097–4.226)/0.026
Dominant model				
C/C	59 (40.1)	55 (32.0)	0.162	1
C/T + T/T	88 (59.9)	117 (68.0)		1.493 (0.919–2.428)/0.106
rs12329760–*TMPRSS2*	150	175		
Genotype				
C/C	109 (72.7)	104 (59.4)	0.041	1
C/T	38 (25.3)	65 (37.2)		1.697 (1.038–2.775)/0.035
T/T	3 (2.0)	6 (3.4)		1.960 (0.474–8.103)/0.353
Allele				
C	0.85	0.78	0.021	-
T	0.15	0.22		
Dominant model				
C/C	109 (72.7)	104 (59.4)	0.017	1
C/T + T/T	41 (27.3)	71 (40.6)		1.717 (1.065–2.769)/0.027
Interaction *ACE1*, *IFIH1* and *TMPRSS2*	145	170		
0, 1, 2 or 3 mutated alleles	141 (97.2)	152 (89.4)	0.013	
4, 5, or 6 mutated alleles	4 (2.8)	18 (10.6)		3.859 (1.271–11.714)/0.017
Females	Survivors	Non-survivors		
rs1990760–*IFIH1*	238	69		
Genotype				
C/C	84 (35.3)	24 (34.8)	0.073	1
C/T	113 (47.5)	25 (36.2)		0.717 (0.365–1.409)/0.334
T/T	41 (17.2)	20 (29.0)		1.889 (0.884–4.037)/0.101
Allele				
C	0.59	0.53	0.236	-
T	0.41	0.47		
Recessive model				
C/C+ C/T	197 (82.8)	49 (71.0)	0.047	1
T/T	41 (17.2)	20 (29.0)		2.281 (1.175–4.427)/0.015
Additive model				
C/C	84 (67.2)	24 (54.5)	0.187	1
T/T	41 (32.8)	20 (45.5)		2.065 (0.942–4.528)/0.070
Dominant model				
C/C	84 (35.3)	24 (34.8)	1.000	1
C/T + T/T	154 (64.7)	45 (65.2)		0.999 (0.545–1.833)/0.999
rs12329760–*TMPRSS2*	240	71		
Genotype				
C/C	164 (68.3)	42 (59.2)	0.192	1
C/T	71 (29.6)	25 (35.2)		1.547 (0.850–2.813)/0.153
T/T	5 (2.1)	4 (5.6)		2.769 (0.690–11.108)/0.151
Allele				
C	0.83	0.77	0.110	-
T	0.17	0.23		
Dominant model				
C/C	164 (68.3)	42 (59.2)	0.196	1
C/T + T/T	76 (31.7)	29 (40.8)		1.647 (0.927–2.927)/0.089
rs2304256–*TYK2*	240	71		
Genotype				
C/C	136 (56.7)	41 (57.7)	0.171	1
C/A	94 (39.1)	23 (32.4)		0.867 (0.473–1.589)/0.644
A/A	10 (4.2)	7 (9.9)		2.438 (0.817–7.279)/0.110
Allele				
C	0.76	0.74	0.651	-
A	0.24	0.26		
Recessive model				
C/C+ C/A	230 (95.8)	64 (90.1)	0.076	1
A/A	10 (4.2)	7 (9.9)		2.574 (0.882–7.513)/0.084
Additive model				
C/C	136 (93.2)	41 (85.4)	0.138	1
A/A	10 (6.8)	7 (14.6)		2.433 (0.806–7.343)/0.115
Dominant model				
C/C	136 (56.7)	41 (57.7)	0.980	1
C/A + A/A	104 (43.3)	30 (42.3)		1.024 (0.582–1.803)/0.934
Interaction *IFIH1*, *TMPRSS2* and *TYK2*	235	69		
0, 1, 2 or 3 mutated alleles	221 (94.0)	60 (87.0)	0.089	
4, 5, or 6 mutated alleles	14 (6.0)	9 (13.0)		2.622 (1.026–6.704)/0.044

Data are shown as number (%) or proportion. * *p*-values were calculated using χ^2^ tests. ^†^
*p*-values and OR (95% CI) obtained using logistic regression analyses adjusting for age and ethnicity. Additive and recessive models were not analyzed for polymorphisms with low frequency of the minor allele.

**Table 6 genes-14-00029-t006:** Genotype and allele frequencies of the polymorphisms of interest in patients with COVID-19 after stratification by ethnicity.

Stratification	Group		Unadjusted *p* *	Adjusted OR (95% IC)/*p* ^†^
Non-White	Inpatients	ICU Patients		
rs223675–*IFNAR2*	84	89		
Genotype				
G/G	45 (53.5)	41 (46.1)	0.080	1
G/A	34 (40.5)	33 (37.0)		1.085 (0.569–2.067)/0.804
A/A	5 (6.0)	15 (16.9)		3.357 (1.114–10.109)/0.031
Allele				
G	0.74	0.65	0.065	-
A	0.26	0.35		
Recessive model				
G/G + G/A	79 (94.0)	74 (83.1)	0.045	1
A/A	5 (6.0)	15 (16.9)		3.238 (1.116–9.391)/0.031
Additive model				
G/G	45 (90.0)	41 (73.2)	0.050	1
A/A	5 (10.0)	15 (26.8)		3.375 (1.116–10.208)/0.031
Dominant model				
G/G	45 (53.6)	41 (46.1)	0.404	1
G/A + A/A	39 (46.4)	48 (53.9)		1.377 (0.753–2.517)/0.299
Non-white	Survivors	Non-survivors		
rs1799752/*ACE1*	123	42		
Genotype				
Del/Del	37 (30.1)	11 (26.2)	0.011	‡
Del/Ins	86 (69.9)	28 (66.7)		
Ins/Ins	0 (0.0)	3 (7.1)		
Allele				
Del	0.65	0.60	0.368	-
Ins	0.35	0.40		
Recessive model				
Del/Del + Del/Ins	123 (100.0)	39 (92.9)	0.020	1
Ins/Ins	0 (0.0)	3 (7.1)		14.56 (1.237–495.6)/0.032 *
Additive model				
Del/Del	37 (100.0)	11 (78.6)	0.025	1
Ins/Ins	0 (0.0)	3 (21.4)		14.98 (1.188–537.5)/0.035 *
Dominant model				
Del/Del	37 (30.1)	11 (26.2)	0.777	1
Del/Ins + Ins/Ins	86 (69.9)	31 (73.8)		1.122 (0.496–2.534)/0.783
rs223675–*IFNAR2*	123	44		
Genotype				
G/G	69 (56.1)	16 (36.4)	0.026	1
G/A	44 (35.8)	19 (43.2)		1.997 (0.892–4.472)/0.093
A/A	10 (8.1)	9 (20.4)		3.682 (1.213–11.176)/0.021
Allele				
G	0.74	0.58	0.006	-
A	0.26	0.42		
Recessive model				
G/G + G/A	113 (91.9)	35 (79.5)	0.053	1
A/A	10 (8.1)	9 (20.5)		2.664 (0.951–7.465)/0.062
Additive model				
G/G	69 (87.3)	16 (64.0)	0.020	1
A/A	10 (12.7)	9 (36.0)		3.823 (1.203–12.149)/0.023
Dominant model				
G/G	69 (56.1)	16 (36.4)	0.038	1
G/A + A/A	54 (43.9)	28 (63.6)		2.336 (1.104–4.944)/0.027
rs12329760–*TMPRSS2*	124	44		
Genotype				
C/C	86 (69.4)	23 (52.3)	0.112	1
C/T	34 (27.4)	18 (40.9)		2.050 (0.948–4.435)/0.068
T/T	4 (3.2)	3 (6.8)		3.090 (0.613–15.577)/0.172
Allele				
C	0.83	0.73	0.042	-
T	0.17	0.27		
Dominant model				
C/C	86 (69.4)	23 (52.3)		1
C/T + T/T	38 (30.6)	21 (47.7)	0.064	2.160 (1.031–4.527)/0.041
Interaction *ACE1* and *TMPRSS2*	123	42		
0, 1 or 2 mutated alleles	120 (97.6)	37 (88.1)	0.040	
3 or 4 mutated alleles	3 (2.4)	5 (11.9)		5.854 (1.280–26.777)/0.023
Interaction *ACE1*, *IFNAR2* and *TMPRSS2*	122	42		
0, 1, 2 or 3 mutated alleles	119 (97.5)	36 (85.7)	0.012	
4, 5 or 6 mutated alleles	3 (2.5)	6 (14.3)		6.468 (1.463–28.592)/0.014

Data are shown as number (%) or proportion. ** p*-values were calculated using χ^2^ tests. ^†^ *p*-values and OR (95% CI) obtained using logistic regression analyses adjusting for age and sex. Adjusted OR for *ACE1* polymorphism was calculated using conditional maximum-likelihood estimate (*cMLE*). ‡ Logistic regression could not be performed because the frequency of the Ins/Ins genotype is zero in the inpatients group.

## Data Availability

The data presented in this study are available on request from the corresponding author.

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
