# Peer review of "Polymorphisms in ACE1, TMPRSS2, IFIH1, IFNAR2, and TYK2 Genes Are Associated with Worse Clinical Outcomes in COVID-19"

_genes, 2022, doi:10.3390/genes14010029_

Round 1

Reviewer 1 Report

Overall, it is a well conceived and executed study which attempts to understand the correlation between polymorphisms in TMPRSS2, ACE1, IFNAR2, and IFIH1 genes with clinical outcomes in COVID-19. Following minor points may be considered to be included in the discussion section

1. As stated by the authors (lines 329-332) rs12329760/TMPRSS2 T allele has been shown to be associated with severity of COVID-19 in other populations as well. Thus, for the present study to be justifiable, authors should mention about reported polymorphisms which have been shown to exhibit differential affects across population. This would make the study more novel rather than confirmatory. This is important because in the subsequent lines (333-336) they refer to a meta study wherein the same allele has been shwon to confer protection against COVID-19.  A table might be added for all studied polymorphisms andn their reported impact. 

2. Whether the polymorphims is repsonsible for the observed effects or increased expression or increased expression due to polymorphism is a very intriguing aspect and the same interplay can be represented as a figure. 

3. Ins allele of the rs1799752/ACE1 has been reported to lower ACE1 expression yet it shows an enahnced risk. Are the number of samples in each category suffcient to make any conclusions?

Author Response

Reviewer 1

Overall, it is a well conceived and executed study which attempts to understand the correlation between polymorphisms in TMPRSS2, ACE1, IFNAR2, and IFIH1 genes with clinical outcomes in COVID-19. Following minor points may be considered to be included in the discussion section

  1. As stated by the authors (lines 329-332) rs12329760/TMPRSS2 T allele has been shown to be associated with severity of COVID-19 in other populations as well. Thus, for the present study to be justifiable, authors should mention about reported polymorphisms which have been shown to exhibit differential affects across population. This would make the study more novel rather than confirmatory. This is important because in the subsequent lines (333-336) they refer to a meta study wherein the same allele has been shown to confer protection against COVID-19.  A table might be added for all studied polymorphisms and their reported impact. 

Answer: Thank you for your comment. We now added a Table showing the reported impact of all studied polymorphisms in the COVID-19 context (Table 1, pages 4-6).

  1. Whether the polymorphism is responsible for the observed effects or increased expression or increased expression due to polymorphism is a very intriguing aspect and the same interplay can be represented as a figure. 

Answer: Thank you for your suggestion. We decided to include this information in the new Table 1, which describes the polymorphisms and reports how they impact COVID-19 susceptibility and gene expression (Table 1, pages 4-6).

  1. Ins allele of the rs1799752/ACE1 has been reported to lower ACE1 expression yet it shows an enhanced risk. Are the number of samples in each category sufficient to make any conclusions?

Answer: Although the Ins allele has been reported to decreased ACE1 expression, we believe that our results showing that the Ins/Ins genotype was associated with risk of ICU admission in females is biologically plausible since there are conflicting results in the literature regarding the association of this polymorphism with COVID-19 (see the new Table 1). For example, although the meta-analysis by Dobrijevic et al showed the Ins/Ins genotype was associated with lower severity, it also showed that the presence of this allele increased the risk of death due COVID-19. Thus, differences in definition of COVID-19 severity as well as ethnicity and sex differences may explain the conflicting results. Moreover, the effect of the Ins allele on ACE1 expression is not well known in COVID-19 patients and its effect on the disease may vary according to interactions with other polymorphisms.

Dobrijevic Z, Robajac D, Gligorijevic N, Sunderic M, Penezic A, Miljus G, et al. The association of ACE1, ACE2, TMPRSS2, IFITM3 and VDR polymorphisms with COVID-19 severity: A systematic review and meta-analysis. EXCLI journal. 2022;21:818-39.

Reviewer 2 Report

Dear Authors, 

please find my comments on the manuscript below:

1. Ad. Supplementary Tables 1, 2, 3, 4 

Reporting of rs368234815 – IFNL4 should be unified. There is a double notation of genotypes and alleles: TT/G, G/G, G or TT/∆G, âˆ†G/∆G, âˆ†G. 

2. Ad. 2.4 Statistical analysis, lines 51 and 52

You mentioned that variables with significant association with COVID-19 in the univariate analysis or biologically relevant were included in the multivariate model, but in my opinion, it is too general information. In the Results section, you presented OR with corresponding 95%CI and p-values for the models including one genetic variable (one specific polymorphism), age and ethnicity (Table 4, Supplementary Tables 1 and 2) or one genetic variable (one specific polymorphism), age and gender (Table 5, Supplementary Tables 3 and 4). In the Statistical analysis subsection, it should be clarified which variables were included in the multivariate models. Information about this included in the footnote of the tables could be overlooked by readers.

3. Ad. 3. Results/ 3.2. Genotype and allele frequencies, lines 198-200: 

The sentence: "The C/T genotype of the rs2285666/ACE2 was more frequent in inpatients compared to ICU patients (22.9% vs. 14.3%, P = 0.007); however, this association was not independent of age, gender, and ethnicity" should be clarified. There is no data about the results of the analysis of the polymorphisms in the context of age, gender, or ethnicity presented in Table 3. 

4. Ad. 3. Results/ 3.2. Genotype and allele frequencies: 

Data about the agreement of genotype distribution with the HWE should be presented (maybe as an additional supplementary table). The short information included in the footnote of Table 3 is not enough to assess the magnitude of the inconsistency quickly. 

5. 3.3 Analyses after stratification by gender and 3.4 Analyses after stratification by ethnicity

Describing the meaning of OR values is confounding in general.  

For example, in lines 235 -238 you mentioned that the Ins/Ins genotype was also more frequent in ICU 236 patients compared to inpatients (3.5% vs. 0.7%, P = 0.002), and the Ins allele was associated with risk for severe COVID-19 (ICU admission) under the dominant model and adjusting for age and ethnicity (OR = 2.283, 95% CI 1.366 – 3.814, P = 0.002).

In fact, the presence of Del/Ins or Ins/Ins genotype in a person (but not the Ins allele pre se) is associated with the increased risk of a severe course of COVID-19 disease. In this case, it is a more than 2 times higher risk in comparison to Del/Del. Moreover, increased/decreased risk should not be reported for alleles because it is calculated for human beings who are carriers of genotypes. 

Best regards

Author Response

Reviewer 2

Dear Authors, 

please find my comments on the manuscript below:

  1. Ad. Supplementary Tables 1, 2, 3, 4 

Reporting of rs368234815 – IFNL4 should be unified. There is a double notation of genotypes and alleles: TT/G, G/G, G or TT/∆G, âˆ†G/∆G, âˆ†G. 

Answer: We corrected this information in the Supplementary Tables.

  1. Ad. 2.4 Statistical analysis, lines 51 and 52

You mentioned that variables with significant association with COVID-19 in the univariate analysis or biologically relevant were included in the multivariate model, but in my opinion, it is too general information. In the Results section, you presented OR with corresponding 95%CI and p-values for the models including one genetic variable (one specific polymorphism), age and ethnicity (Table 4, Supplementary Tables 1 and 2) or one genetic variable (one specific polymorphism), age and gender (Table 5, Supplementary Tables 3 and 4). In the Statistical analysis subsection, it should be clarified which variables were included in the multivariate models. Information about this included in the footnote of the tables could be overlooked by readers.

Answer: Thank you for your comment. We added the information about which variables were included in the multivariate models in the Statistical Analysis Section (page 7).

  1. Ad. 3. Results/ 3.2. Genotype and allele frequencies, lines 198-200: 

The sentence: "The C/T genotype of the rs2285666/ACE2 was more frequent in inpatients compared to ICU patients (22.9% vs. 14.3%, P = 0.007); however, this association was not independent of age, gender, and ethnicity" should be clarified. There is no data about the results of the analysis of the polymorphisms in the context of age, gender, or ethnicity presented in Table 3. 

Answer: We now changed this sentence to clarify this information (page 13) and also included a new Supplementary Table with the results of logistic regression analysis for data presented in the old Table 3 (now Table 4) (Supplementary Table 1). We opted to not include in Table 4 because this table is already too long.

  1. Ad. 3. Results/ 3.2. Genotype and allele frequencies: 

Data about the agreement of genotype distribution with the HWE should be presented (maybe as an additional supplementary table). The short information included in the footnote of Table 3 is not enough to assess the magnitude of the inconsistency quickly. 

Answer: Thank you for your comment. We now added the information about HWE in the old Table 3 (now Table 4).

  1. 3.3 Analyses after stratification by gender and 3.4 Analyses after stratification by ethnicity

Describing the meaning of OR values is confounding in general.  

For example, in lines 235 -238 you mentioned that the Ins/Ins genotype was also more frequent in ICU 236 patients compared to inpatients (3.5% vs. 0.7%, P = 0.002), and the Ins allele was associated with risk for severe COVID-19 (ICU admission) under the dominant model and adjusting for age and ethnicity (OR = 2.283, 95% CI 1.366 – 3.814, P = 0.002).

In fact, the presence of Del/Ins or Ins/Ins genotype in a person (but not the Ins allele per se) is associated with the increased risk of a severe course of COVID-19 disease. In this case, it is a more than 2 times higher risk in comparison to Del/Del. Moreover, increased/decreased risk should not be reported for alleles because it is calculated for human beings who are carriers of genotypes. 

Answer: Thank you for your comments. Please note that we did not report risk for alleles, but for carriers of these alleles, as frequently used in genetic association papers. For example, presence of the risk allele T means those patients carrying this allele (dominant model). However, we clarified this information in the results (Results Section, pages 22 and 27).

Best regards

Reviewer 3 Report

In this study, the authors investigated 9 SNPs in the ACE1, ACE2, DPP9, IFIH1, IFNAR2, IFNL4, TLR3, TMPRSS2, and TYK2 genes and the severity and mortality in COVID-19 patients. Although this topic is interesting and the study is well-designed, I have several concerns that should be addressed, as follows:

  • Introduction:
  • The authors should mention the location of the investigated genes.
  • The roles of the DPP9 and TYK2 genes in COVID-19 should be defined in more detail.
  • Methods:
    • The patients were divided according to COVID-19 severity. Please define the severity criteria and provide references.
    • 2.3 Genotyping: Please include the primers’ sequences used in genotyping in a table.
  • Results:
    • The results of different genetic models for all investigated genes should be tabulated with OR, CI, and p-values. I recommend revising Table 3 and dividing it into two tables: one for severity and one for mortality, with the addition of all genetic models and the risk ratio.
    • The paragraph (Lines 233–246) should be revised for accuracy as it contains several data that are not matched with those illustrated in Table 3.
  • Throughout the manuscript, there are several structural, grammatical, and capitalization errors. The whole manuscript should be carefully revised by a native English speaker.
  • Abbreviations should be defined in full terms when they are first used in the text, such as STROBE, STREGA, HPLC, LDH, and PCR.

Author Response

Reviewer 3

In this study, the authors investigated 9 SNPs in the ACE1, ACE2, DPP9, IFIH1, IFNAR2, IFNL4, TLR3, TMPRSS2, and TYK2 genes and the severity and mortality in COVID-19 patients. Although this topic is interesting and the study is well-designed, I have several concerns that should be addressed, as follows:

  • Introduction:
  • The authors should mention the location of the investigated genes.

Answer: We added now this information in the new Table 1.

  • The roles of theDPP9 and TYK2 genes in COVID-19 should be defined in more detail.

Answer: Thank you for your comment. We now added more information about DPP9 and TYK2 in the Introduction Section (page 2).

  • Methods:
    • The patients were divided according to COVID-19 severity. Please define the severity criteria and provide references.

Answer: Hospitalized patients with COVID-19 were separated according to disease severity. Severity criteria were defined as the need to ICU admission, and patients were categorized as critically ill patients with severe COVID-19 (ICU patients, n=414) or non-critically ill patients (ward inpatients, n=280). All critically ill patients were admitted to the ICU. At our hospital, during pandemic, only patients in acute respiratory failure were admitted to the ICU, with the following criteria: respiratory frequency >30 rpm, oxygen saturation <90%, or the need for invasive or non-invasive mechanical ventilation, same as suggested in literature (Hajjar, LA. 2021). Non-critically ill patients did not fulfil the above criteria and were admitted to the medical ward, where they did not receive any kind of ventilation support besides supplemental oxygen. This information was better detailed in the Materials and Methods (pages 2 and 3). A new reference was added to References.

Hajjar, L.A., Costa, I.B.S., Rizk, S.I. et al. Intensive care management of patients with COVID-19: a practical approach. Ann. Intensive Care 11, 36 (2021). https://doi.org/10.1186/s13613-021-00820-w

2.3 Genotyping: Please include the primers’ sequences used in genotyping in a table.

Answer: Polymorphisms were genotyped using specific Human TaqMan SNP Genotyping Assays 40x (Thermo Fisher Scientific), which consists of two sequence-specific primers and two TaqMan minor groove binder (MGB) probes with non-fluorescent quenchers (NFQ), one probe for each allele. Having the Assay ID of each polymorphism you can search the specific sequences in the Thermo Fisher Scientific site (https://www.thermofisher.com/br/en/home.html ). Thus, we opted to not include sequences in the manuscript, but added the assay IDs in the Material and Methods and how to obtain their sequences in the mentioned site (page 3).

  • Results:
    • The results of different genetic models for all investigated genes should be tabulated with OR, CI, and p-values. I recommend revising Table 3 and dividing it into two tables: one for severity and one for mortality, with the addition of all genetic models and the risk ratio.

Answer: Thank you for your suggestion. As we already have six tables in the article and Table 3 is already too long; so, we decided to include the results of logistic regression for data present in the old Table 3 (now Table 4) in a new Supplementary Table 1.

  • The paragraph (Lines 233–246) should be revised for accuracy as it contains several data that are not matched with those illustrated in Table 3.

Answer: The information in this paragraph (lines 233-246) are about the data illustrated in new Table 4 (old Table 3), as we described in lines 220 and 221 “In women, some of the analyzed polymorphisms were associated 221 with COVID-19 outcomes (Table 4) and their results are detailed below”.

  • Throughout the manuscript, there are several structural, grammatical, and capitalization errors. The whole manuscript should be carefully revised by a native English speaker.

Answer: The article was sent for English revision in a specialized company.

  • Abbreviations should be defined in full terms when they are first used in the text, such as STROBE, STREGA, HPLC, LDH, and PCR.

Answer: We now defined the abbreviations.

Round 2

Reviewer 1 Report

The authors have satisfactorily answered all the raised queries. The manuscript can now be accepted for publication. 

Author Response

Thank you for your comment. 

Reviewer 2 Report

Dear Authors, 

I have no additional comments on the manuscript.

Bests regards 

Author Response

Thank you for your comment. 

Reviewer 3 Report

  • The authors have adequately addressed most of my concerns; however, one concern remains to be addressed:
  • - The paragraph (Lines 267–279) should be revised for accuracy as it contains several data that are not matched with those illustrated in Table 4.

Author Response

The information in this paragraph (lines 267-279) is related to the data illustrated in Table 5, as we described in lines 249 and 254 “Table 5 and Supplementary Tables 2 and 3 show frequencies of the analyzed  polymorphisms in patients with COVID-19, stratified by sex. In males, no significant  difference was found in the distributions of the 9 polymorphisms of interest between inpatients and ICU patients or between survivors and non-survivors (all P > 0.050, Supplementary Table 3). In females, some of the analyzed polymorphisms were associated with COVID-19 outcomes (Table 5) and their results are detailed below. ”. We had identified a wrong table in the reviewer response, but the information in lines 267-279 is correct in the article.

Round 3

Reviewer 3 Report

The authors have adequately addressed all my concerns.